# Influence of Anesthesia and Clinical Variables on the Firing Rate, Coefficient of Variation and Multi-Unit Activity of the Subthalamic Nucleus in Patients with Parkinson’s Disease

**DOI:** 10.3390/jcm9041229

**Published:** 2020-04-24

**Authors:** Michael J. Bos, Ana Maria Alzate Sanchez, Raffaella Bancone, Yasin Temel, Bianca T.A. de Greef, Anthony R. Absalom, Erik D. Gommer, Vivianne H.J.M. van Kranen-Mastenbroek, Wolfgang F. Buhre, Mark J. Roberts, Marcus L.F. Janssen

**Affiliations:** 1Department of Anesthesiology and Pain Medicine, Maastricht University Medical Center, P. Debyelaan 25, 6229 HX Maastricht, The Netherlands; wolfgang.buhre@mumc.nl; 2School for Mental Health and Neuroscience, Faculty of Health, Medicine and Life Sciences, Maastricht University, Universiteitssingel 40, 6229 ER Maastricht, The Netherlands; a.alzatesanchez@alumni.maastrichtuniversity.nl (A.M.A.S.); raffaellabancone@gmail.com (R.B.); y.temel@mumc.nl (Y.T.); bianca.greef@mumc.nl (B.T.A.d.G.); e.gommer@mumc.nl (E.D.G.); v.kranen.mastenbroek@mumc.nl (V.H.J.M.v.K.-M.);; 3Department of Neurosurgery, Maastricht University Medical Center, P. Debyelaan 25, 6229 HX Maastricht, The Netherlands; 4Department of Neurology, Maastricht University Medical Center, P. Debyelaan 25, 6229 HX, Maastricht, The Netherlands; 5Department of Anesthesiology, Groningen University, University Medical Center Groningen, Hanzeplein 1, 9713 GZ Groningen, The Netherlands; a.r.absalom@umcg.nl; 6Department of Clinical Neurophysiology, Maastricht University Medical Center, P. Debyelaan 25, 6229 HX Maastricht, The Netherlands; 7Faculty of Psychology and Neuroscience, Maastricht University, Universiteitssingel 40, 6229 ER Maastricht, The Netherlands; mark.roberts@maastrichtuniversity.nl

**Keywords:** deep brain stimulation, microelectrode recordings, subthalamic nucleus, procedural sedation and analgesia, Parkinson’s disease, clonidine, dexmedetomidine, remifentanil

## Abstract

Background: Microelectrode recordings (MER) are used to optimize lead placement during subthalamic nucleus deep brain stimulation (STN-DBS). To obtain reliable MER, surgery is usually performed while patients are awake. Procedural sedation and analgesia (PSA) is often desirable to improve patient comfort, anxiolysis and pain relief. The effect of these agents on MER are largely unknown. The objective of this study was to determine the effects of commonly used PSA agents, dexmedetomidine, clonidine and remifentanil and patient characteristics on MER during DBS surgery. Methods: Data from 78 patients with Parkinson’s disease (PD) who underwent STN-DBS surgery were retrospectively reviewed. The procedures were performed under local anesthesia or under PSA with dexmedetomidine, clonidine or remifentanil. In total, 4082 sites with multi-unit activity (MUA) and 588 with single units were acquired. Single unit firing rates and coefficient of variation (CV), and MUA total power were compared between patient groups. Results: We observed a significant reduction in MUA, an increase of the CV and a trend for reduced firing rate by dexmedetomidine. The effect of dexmedetomidine was dose-dependent for all measures. Remifentanil had no effect on the firing rate but was associated with a significant increase in CV and a decrease in MUA. Clonidine showed no significant effect on firing rate, CV or MUA. In addition to anesthetic effects, MUA and CV were also influenced by patient-dependent variables. Conclusion: Our results showed that PSA influenced neuronal properties in the STN and the dexmedetomidine (DEX) effect was dose-dependent. In addition, patient-dependent characteristics also influenced MER.

## 1. Introduction

Deep brain stimulation (DBS) of the subthalamic nucleus (STN) is a well-established procedure for the treatment of refractory Parkinson’s disease (PD) [1]. The clinical outcome of the surgery largely depends on correct positioning of the stimulating electrode in the sensorimotor region of the STN [2,3]. Microelectrode recordings (MER) of single-cell and multi-unit neuronal activity are commonly used to verify the borders of the STN [4]. To obtain reliable MER, DBS surgery is traditionally performed under local anesthesia alone, as the sedative and anesthetic agents may interfere with neural activity. However, patients may experience pain, anxiety or other forms of discomfort. To improve patient comfort and tolerance of the DBS implantation procedure, procedural sedation and/or analgesia (PSA) may be applied [5].

Propofol is commonly used for sedation during DBS implantation. It exerts its clinical effect through an agonist effect on the gamma-aminobutyric acid type A (GABA_A_) receptor [6]. Several studies have shown that GABAergic agents alter STN activity in a dose-dependent manner [7,8,9]. In the past decade experience has been gained with non-GABA-mediated agents, including the α_2_-agonists clonidine (CLONI) and dexmedetomidine (DEX), which possess sedative and mild analgesic effects, and the ultrashort acting opioid, remifentanil (REMI) which provides potent analgesia and mild sedation. Since the pharmacokinetic effects of these drugs are not mediated by GABA_A_ receptors, their influence on STN neuronal activity is postulated to be less pronounced than of GABAergic agents. However, the available literature on non-GABA-mediated PSA effects on STN neuronal activity is sparse and consists largely of small uncontrolled retrospective case series with poor control over heterogeneity in patient cohorts [10,11,12,13,14,15,16].

The primary aim of this study was to determine the effect of the non-GABAergic PSA agents DEX, CLONI and REMI on MER in patients with PD during DBS electrode implantation surgery. In addition to PSA-related effects on MER, we analyzed patient characteristics such as age, disease severity and disease duration to control for heterogeneity in the sample.

## 2. Materials and Methods

### 2.1. Subjects

After gaining the approval of the local ethical committee (METC Maastricht University Medical Center, the Netherlands, protocol number 184214) we conducted a retrospective analysis of data from all PD patients who underwent DBS surgery in Maastricht UMC+ between January 2009 and December 2018. We acquired patient demographics and anesthetic data and retrieved the raw MER data for offline processing and analysis.

### 2.2. Anesthetic Management

All patients underwent a multidisciplinary preoperative assessment of eligibility for DBS surgery. In the operating room, standard monitoring was applied including a five-lead electrocardiogram, pulse oximetry, inspiratory and expiratory O_2_ and CO_2_ monitoring and invasive blood pressure monitoring. DBS surgery was performed under local anesthesia alone or in combination with PSA administered at the discretion of the responsible anesthesiologist. The goal of PSA was to maintain mild to moderate sedation, with the patient responsive to verbal command (so-called conscious sedation). The skin puncture sites of the stereotactic frame pins, and the surgical incision sites, were infiltrated with a 50:50 mixture of lidocaine 1% and levobupivacaine 0.5% with epinephrine (1:100.000). During the procedure some patients received no sedative drugs, whereas some received one or more of DEX, CLONI or REMI by continuous intravenous infusion for PSA. Some patients received DEX only in the first phase of the surgery until around 20 min before the start of MER (Table 1). After DBS electrode implantation, all patients underwent general anesthesia for tunneling of the extension cables and placement of the pulse generator. Postoperatively, patients were transferred to the post-anesthesia care unit for hemodynamic- and neuro-monitoring.

### 2.3. Surgical Procedure

After placement of a Leksell stereotactic frame (Elekta, Stockholm, Sweden), a computed tomography (CT) scan of the head was performed. The CT-image was co-registered with a magnetic resonance imaging (MRI) that had been performed before surgery. Following target identification and trajectory planning, a burr hole was drilled, micro-electrodes (model 230766, Medizintechnik GmbH, Emmendingen, Germany) were implanted and recordings were performed. Visual and auditory confirmation of the target was performed by a neurophysiologist. Then, macrostimulation and neurological testing were carried out. The presence of good quality electrophysiological recordings and few or no side effects indicated the optimal contact point at which a quadripolar electrode (model 3387 or 3389; Medtronic, Minneapolis, USA) was placed. A postoperative CT scan was performed within 24 h in order to exclude intracerebral hemorrhages and to evaluate the final position of the electrodes.

### 2.4. MER Acquisition

Up to 5 microelectrodes were used to record neuronal activity along the planned trajectory in order to identify the target. Data were recorded from 10 mm above to 5 to 7 mm below the target in steps of 0.5–1 mm for approximately 30 s at each recording location (mean 35.94 s, SD 15.93). Data from a typical electrode track is shown in Figure 1A.

The target for all patients was the STN. The electrodes classically passed through the thalamus, zona incerta, STN and sometimes reached into the dorsal border of the substantia nigra reticulata. The electrode signal was sampled at 20 or 25 kHz, bandpass filtered online, (V3.15; Inomed Medizintechnik GmbH, Emmendingen, Germany) and saved for offline analysis. The first 41 patients were recorded with a high-pass filter 160 Hz, thereafter the high-pass filter was at 0 Hz, low-pass filter was at 5000 Hz.

### 2.5. Data Processing and Analysis

Custom-written MATLAB scripts (V2017B; MathWorks) were used to conduct the data-analysis. For each recording, the raw data were high-pass filtered at 300 Hz prior to visual and auditory inspection. To identify single unit (SU) activity, periods of interest were manually selected to exclude periods of high noise or unstable SU activity. Spike times were identified by signal crossings of a manually set threshold (Figure 1B). Spikes representing SUs were selected using principal component analysis and K means clustering (Figure 1C). Manual selection was used for sorting SU clusters. SU clusters were confirmed as SUs by inspecting their autocorrelation, with a minimum gap of 2 ms between spikes representing the refractory period. For added robustness, this analysis was independently performed by 4 authors (A.M.A.S., M.J.B., R.B. and M.J.R.) (Figure 1C–G). For the main analysis data sorted by A.M.A.S. was used, who inspected all recording sites in our sample. Firing rate (spikes per second) of the SUs were calculated by dividing the number of spikes by the recording time. The coefficient of variation (CV) was defined by dividing the standard deviation of the inter-spike interval by the mean. 

To identify multi-unit activity (MUA) we calculated the power-spectral density of the signal within the bandwidth of 100 and 500 Hz. Periods of high noise were automatically identified and rejected by calculating the root-mean squared (RMS) of the high-pass filtered data in segments of 50 ms. Periods in which the RMS exceeded the median RMS + 3 standard deviations were excluded from further analysis [17]. Following this procedure, the surviving raw (unfiltered) data were cut into non-overlapping snips of 250 ms and the power-spectral density was calculated using a multitaper method with discrete prolate spheroidal sequences using the Fieldtrip MATLAB toolbox [18]. To account for non-biological differences between recording tracts (electrode and tissue impedance etc.) power at each frequency was expressed as a ratio with respect to power at the first 5 recording sites. Finally, MUA total power, hereafter referred to simply as MUA, was calculated as the sum of all baseline corrected power above 300 Hz.

### 2.6. Statistical Analysis

Statistical analyses was performed using custom-written MATLAB scripts (V2017B; MathWorks) to test the effect of PSA agents and other variables on electrophysiological measures. For all tests, the threshold for statistical significance was set to α = 0.05. Patients were divided into 7 groups according to their sedative administration (no PSA (control group)), PSA discontinued (discontinued before MER), DEX, REMI, CLONI, DEX and REMI, CLONI and REMI (Table 1)).

As a first step of the statistical analysis, one-way ANOVAs were conducted for each electrophysiological measure to test for differences between groups. This was followed by multiple two-sample *t*-tests comparing data from each PSA group with data from the no PSA group. *P*-values were corrected for multiple comparison using the Benjamini & Hochberg method for control of the false discovery rate (FDR) [19].

For further insight, linear regression analysis was conducted using the applied PSA drugs, and clinical and demographic variables as predictors to test their effect on firing rate, CV and MUA. Additional linear regression analysis with a random effect grouped by patient ID were conducted, considering the clustered nature of the data.

Finally, we tested the effect of the PSA dose. We focused on DEX since the dose of CLONI was not consistent, which made impossible to run a dose-dependent analysis, while the effect of REMI was small. We conducted a correlation analysis including data from patients who received either DEX alone, or a combination of DEX and REMI. A standard linear model (no random effects) was constructed.

## 3. Results

### 3.1. Demographics

From January 2009 to December 2018 a total of 93 PD patients underwent STN-DBS insertion. Data from 13 patients were excluded from analysis because of incomplete data. Two patients underwent surgery under general anesthesia and were also excluded from further analysis. Anesthetic and electrophysiological data from the remaining 78 patients were analyzed, thus yielding electrophysiological date from 156 cerebral hemispheres (Table 1).

Data were first grouped into clusters depending on whether data were acquired when patients were awake or sedated. The awake cluster included data from two patient groups: the first group received no sedatives and the second group of patients received DEX or a combination of DEX and REMI which was discontinued approximately 20 min before MER. The data in the sedation cluster was sub-divided according to the PSA applied during surgery: DEX, CLONI, REMI, or a combination of DEX and REMI or REMI and CLONI (Table 1).

To control for systematic differences between groups, we tested whether demographic characteristics (age, disease duration, Unified Parkinson Disease Rating Scale (UPDRS), side of onset, sex and weight) were equivalent across groups. One-way ANOVA showed no difference for any of these variables (respectively F(6,71) = 1.14, *p* = 0.35; F(6,71) = 0.59, *p* = 0.74; F(6,71) = 0.40, *p* = 0.88; F(6,71) = 0.39, *p* = 0.88; F(6,71) = 0.46, *p* = 0.83, F(6,71) = 0.39, *p* = 0.88 no correction for multiple comparisons were applied since we were here more concerned with minimizing type 2 errors).

### 3.2. Discontinuation of Sedative Agents

In 7 patients, PSA agents (DEX *n* = 4, DEX and REMI *n* = 3) were given during the first part of the operation and were discontinued approximately 20 min before the start of MER. While these patients appeared clinically to be awake during surgery, it is possible that these compounds might still have affected the electrophysiological properties of the STN even after their discontinuation [14]. Therefore, we tested whether the group who received no PSA drugs and the group of patients in whom PSA was discontinued should be considered as distinct groups for further analysis. We conducted a *t*-test for comparison of the electrophysiological measures (firing rate (sp/sec), CV and MUA) between these groups. In SU activity there was a trend towards decreased firing rate (*t*(214) = 1.63, *p* = 0.11), but no difference in the CV (*t*(214) = −0.18, *p* = 0.86). MUA was significantly reduced in the discontinued drug group (*t*(1571) = 5.67, *p* < 0.0001). Given these findings (Figure 2) we considered the patients in whom PSA agents were discontinued as a separate group from patients who received no PSA for further analysis.

### 3.3. Effect of PSA Agents on Firing Rate, Coefficient of Variation and Multi-Unit Activity

A one-way ANOVA was conducted between the seven groups for each electrophysiological measure. Significant, or trending differences between groups for all measures were observed (Firing rate, F(6,581) =2.08, *p* < 0.054; CV, F(6,581) = 3.69, *p* < 0.001; MUA, F(6,4088) = 15.93, *p* < 0.0001). To further test for differences between groups, a post-hoc two-sample *t*-tests was conducted to compare each PSA drug against the control group (*n* = 165) with respect to firing rate, CV and MUA.

No significant differences in the firing rate between groups after FDR correction for multiple comparisons were found, although DEX and REMI, and REMI in combination with CLONI showed a trend towards significance (*t*(209) = 2.64, *p* = 0.0531; *t*(269) = 2.28, *p* = 0.0695).

For CV, there were significant differences between the no drug group and the groups DEX (*t*(249) = −3.84, *p* = 0.00092), REMI (*t*(256) = −2.65, *p* = 0.013), DEX and REMI group (*t*(209) = −3.05, *p* = 0.0078), and REMI and CLONI (*t*(269) = −2.62, *p* = 0.013).

All PSA drug groups except REMI showed a significant decrease in MUA (*n* = 1255, PSA discontinued, *t*(1594) = 4.95, *p* < 0.0001; DEX, *t*(1820) = 6.73, *p* < 0.0001; CLONI, *t*(1594) = 4.45, *p* < 0.0001; DEX and REMI, *t*(1522) = 3.94, *p* < 0.0001; and REMI and CLONI, *t*(1934) = 3.69, *p* < 0.0001), REMI (*t*(1891) = 1.31, *p* = 0.19). All p values were corrected for multiple comparison using FDR (Figure 3).

To account for variance in the patient groups, we conducted a linear regression analysis to test the effect of the PSA agents with clinical and demographic variables included as factors. For DEX, CLONI, REMI, sex, recording location with respect to onset side of the disease, and left or right hemisphere were defined as categorical variables. We further included age at surgery (years), weight (Kg), UPDRS III pre-operative off medication, disease duration at surgery (months) and depth of the recording site within the STN (mm from the dorsal border) as continuous variables in the model (Table 2).

There were no significant effects of the PSA agents in the firing rate, but we could observe some trends. Notably, DEX was weakly associated with a reduced firing rate. For CV, DEX showed a significant effect and REMI showed a small effect. DEX and PSA discontinued showed significant effects in the MUA, while CLONI and REMI did not. The slope (estimates) indicated the direction of the effect. In this sense, the effect of the PSA agents in CV indicates an increase, while the slope for MUA indicates a decrease. These findings supported our previous analysis, except CLONI, where effects identified in the *t*-test were not significant in this analysis.

In addition to effects of PSA agents, we found that electrophysiological measures were affected by several clinical and demographic variables. Disease duration showed a negative effect on CV, thus CV decreased as disease advanced. Age showed a significant impact on MUA, thus MUA decreased with increased age. Hemisphere (left or right) was also significantly associated with MUA, thus right hemisphere showed higher power than the left hemisphere. Finally, STN depth had a significant impact on MUA (Table 2).

Taking into consideration that we were working with multiple observations within each patient, we also conducted a linear regression model with a random effect grouped by patient for all three electrophysiological measures. This more conservative analysis brings a reduction in the statistical power, nevertheless for CV, the effect of DEX, REMI and disease duration remained significant. For MUA, the effect of DEX, PSA discontinued, hemisphere and age also remained significant. The non-significant effect on firing rate also remained (Appendix A).

### 3.4. Interrater Reliability

SUs were identified and sorted by hand which is an inevitably subjective process (Figure 1). To test whether the effects we report were robust, every recording was inspected by at least two individuals (every site was inspected A.M.A.S. and one of M.J.B., M.J.R. or R.B.). We repeated the linear regression analysis for SU data (spikes/seconds and CV) using data for units identified by M.J.B., M.J.R. or R.B., including the sorter ID as an additional categorical variable. All main effects we report on were replicated in this analysis (Appendix A).

### 3.5. Dose-Dependent Effect of DEX 

Our analysis showed that DEX had a significant impact on electrophysiological measures. To test whether these effects were dose-dependent, we conducted further analysis focused only on patients who received DEX during surgery. We first performed a simple correlation analysis between each measure and DEX dose (Figure 4). The correlations were significant, but the correlation coefficient was low for all measures. For illustration, we extrapolated the regression line to include 0 dose and plotted the data for the 0 dose patients (not included PSA discontinued group). For both firing rate and CV the regression line passed close to the mean of the 0 dose data, while this was not the case for MUA. This may indicate a non-linear effect of DEX on MUA whereby even a small dose leads to strong suppression. This interpretation would be in line with our finding that discontinuation of DEX, 20 min before recordings MUA also had a large impact. One interesting point is that the analysis of firing rate was significant, showing a decrease in firing rate with increased dose. This might mean that the trend observed in the *t*-test and linear regression analysis is a true effect.

## 4. Discussion

In this retrospective analysis, we studied the influence of the non-GABAergic drugs DEX, CLONI and REMI on the firing rate and CV of single neuron activity, as well as the power of MUA in the STN of PD patients undergoing DBS surgery. The results of the present study demonstrated that, even when PSA was discontinued around 20 min prior to MER, this still had significant effect on MUA. A trend was observed for a reduced firing rate by DEX, which was significant on the correlation analysis. Both DEX and REMI showed an increase in CV, but only DEX showed a decrease in MUA. The effect of DEX was dose-dependent. CLONI showed no effect on all measures. Lastly, several patient-dependent variables, such as age, disease duration and left or right hemisphere influenced MUA and CV.

### 4.1. Effect of Procedural Sedation and Analgesia on Micro-Electrode Recordings

The use of PSA agents and their effects on neuronal activity has been debated since the initiation of DBS surgery. Traditionally, the anesthetic management approach comprised local anesthesia with monitored care to facilitate MER. However, recent work has shown that 40% of patients suffer from pain, severe OFF-symptoms and intolerable exhaustion during the hours of awake surgery [5]. To improve patient acceptance, sedation is thus commonly administered. Propofol, a GABAergic agent, has been most frequently used when sedation is required. As stated before, several clinical reports showed a dose-dependent effect of propofol on STN neuronal activity in patients with PD [7,8,9,20]. Propofol reduces neuronal activity by enhancing inhibitory neurotransmission and reducing excitatory neurotransmission [21]. Although some studies showed good quality MER with low-dose propofol, potent GABAergic agents such as propofol should not be the first choice for PSA during DBS surgery [22,23,24].

Alternatively, α_2_-agonists (non-GABAergic) are useful in this regard. Currently, two different α_2_-agonists are commonly used in clinical practice: CLONI and DEX. While both drugs have anxiolytic, sedative and analgesic properties, DEX is a more selective α_2_-receptor agonist than CLONI. Since central α_1_-receptor activation counteracts sedative α_2_ effects, DEX has a more profound sedative effect [25]. The sedative effects of α_2_-agonists are mediated through activation of pre- and postsynaptic α_2_-receptors in the locus coeruleus which has noradrenergic afferent connections with the STN [26,27]. This route provides a plausible mechanism for the effects observed in the current study, since it has been shown that noradrenergic modulation with α_1_ and α_2_-agonists change firing rate and firing patterns of STN neurons, in line with our findings [28,29].

#### 4.1.1. Dexmedetomidine

In this study we analyzed quantitative effects of DEX on MER following two different PSA protocols. In the first protocol PSA agents (DEX alone or DEX and REMI) were discontinued approximately 20 min before the start of MER. It can be speculated that the effects on MER in the discontinued group are solely DEX effects, since REMI has a very short context-sensitive half-time of 3–4 min. In these patients, MUA was significantly suppressed while, SU activity showed a trend towards a lower firing rate but no change in CV. Interestingly, an earlier study by Mathews et al., in which a similar PSA protocol was used (discontinuation of PSA agents before the MER), reported no difference in firing rate, but showed a significant decrease in CV [14]. Their protocol was not identical to ours. Patients received either REMI in bolus (the control group in that study), or DEX and REMI in bolus prior to MER. Moreover, the dose of DEX was higher in their study (0.1–1.0 µg kg^−1^ h^−1^ versus 0.2–0.5 µg kg^−1^ h^−1^). In another case series by Kwon et al., patients received a loading dose of DEX 0.9 µg kg^−1^ followed by a maintenance dose of 0.5 µg kg^−1^ h^−1^ combined with REMI at 0.05 µg kg^−1^ min^−1^ and propofol was administered in small boluses. Using this protocol, the depth of sedation was maintained at a level of slight sedation (Bispectral index (BIS) of 80). All PSA agents were discontinued 20 min before MER. In that study, firing rates of STN neurons were significantly reduced compared to the control group who received no sedatives [15]. The suppression of firing rates in this protocol (that included higher DEX dose in addition to the high loading does) is in line with our finding of a dose-dependent suppression, although it is challenging to compare previous studies in which DEX was discontinued before MER with our findings, due to different dose regimens. Taken together, the previous literature and our data suggest that DEX still has an effect on MER after 20 min of discontinuation.

In our study, a second group of patients received continuous DEX infusions, alone or in combination with REMI during MER. Firing rates were not significantly altered, but there was a trend to lowering. Both CV and MUA were significantly affected whereby CV was increased and MUA decreased. Correlation analysis showed that these effects were dose-dependent, including lowering of the firing rate. A small case series reported a suppression of neuronal activity in patients who received DEX sedation throughout the full procedure (range 0.1–0.4 µg kg^−1^ h^−1^) in comparison to patients in which DEX was discontinued before recordings [12]. In another study, patients received a bolus DEX of 0.5–1.0 µg kg^−1^ followed by a maintenance infusion of 0.1–1.0 µg kg^−1^ h^−1^. They reported a slight increase in firing rate and a significant decrease in burst index (decreased number of spikes within a burst) compared to patients who received no sedation [13]. Thus, these findings appear contradictory to our findings as well as with previous literature. A possible explanation for the differences with our findings is the dose they used, which included a bolus followed by a relatively high maintenance dose.

To summarize our results, dexmedetomidine caused a trend toward decreased firing rates, significantly suppressed MUA and increased CV. These effects were present even at low dose and even after discontinuation of DEX. However, direct comparison of our findings with previous studies is challenging, because of the variability of the sedation protocols, patients groups and methodology.

#### 4.1.2. Clonidine

CLONI is a α_2_-receptor agonist with a pharmacodynamic profile almost identical to DEX but is less selective for the α_2_-receptor than DEX. As a consequence, effects on the locus coeruleus are less profound and therefore CLONI would be expected to have less impact on STN neural firing compared to DEX. To our knowledge, no studies have yet reported effects of CLONI on MER during DBS surgery.

In our study, patients received CLONI alone or in combination with REMI. Compared to the no PSA group, these patients showed no significant differences in neuronal firing rates, CV or MUA, although firing rates and MUA showed a trend towards a decrease. Moreover, in our linear regression analysis the effect estimate associated with CLONI was consistently lower than the estimate associated with DEX. Thus, in line with our expectation, CLONI appeared to show a similar but less profound effect on STN neurons compared to DEX. It should be noted however, that direct comparison between the two agents is complex because of the difference in pharmacokinetic profiles. In our study the comparison was further limited due to the heterogeneity in CLONI doses and the lower number of patients in the CLONI group.

#### 4.1.3. Remifentanil

The last PSA agent we investigated was remifentanil. Remifentanil is an ultrashort-acting µ-receptor opioid agonist with a rapid time to peak activity after a bolus dose and a short context-sensitive half-time of less than 4 min without regard to infusion duration [30]. In rats, opioids have been reported to exert an inhibitory modulation of GABAergic and glutamatergic synaptic transmission in the STN via presynaptic µ- and δ-receptors [31,32]. Only a limited number of studies have addressed effects of opioids during STN-DBS surgery [9,22,23,33]. In these studies, neurophysiological data were obtained while patients received propofol in combination with opioids. Therefore, the opioid effects on neuronal activity could not be well characterized. To our knowledge only one other study has assessed the effect of REMI on STN neurons in PD patients [24]. In that study, single cell activity of 4 neurons were analyzed before and after a bolus of 0.5 µg kg^−1^. REMI did not significantly alter short interval discharge activity, a measure related to firing rate. Our results also showed no significant effect of REMI on firing rates or MUA. The use of REMI was only associated with a significant increase in CV compared to the control group.

### 4.2. Effect of Clinical and Demographic Variables on Microelectrode Recordings

In addition to the effects of the PSA agents, clinical and demographic variables might affect neuronal firing properties of the STN. Therefore, we tested the effect of these variables on electrophysiological recordings.

One interesting finding was that the CV increased in patients with a longer disease duration. An increase in burst activity is generally observed in animal models of PD [34,35]. One potential explanation of this phenomenon is that neuronal firing becomes more strongly locked to a more powerful beta rhythm, seen with advanced disease, thereby reducing variability in the interspike-interval. The underlying mechanism of our finding therefore remains elusive. Finally, while CV was related to disease duration, no relation with the UPDRS III score, side of onset or age of the patients was observed.

The hemisphere of the recordings significantly influenced MUA with the right STN having higher power than the left. Currently no study to our knowledge has reported lateralization of MUA in the STN during rest. Interestingly, several studies in humans and animals have shown that the left and right cortex are related to different functions. In line with functional lateralization in the cortex, emotions seems to be processed in the right STN [36,37,38,39]. We therefore speculate that the difference we observed may be related to functional lateralization of the STN; a hypothesis that needs further validation.

Finally, the age of the patient significantly influenced MUA. Within the STN, older patients showed lower power compared to younger patients. An effect of aging on neuronal power spectrum in the cortex has been reported previously [40,41,42]. In contrast to our findings, these studies all reported an increase in high frequency power with age, which they discuss in terms of the neuronal noise hypothesis of aging [43]. A number of interesting hypothesis may account for the reversal of this pattern that we observed in STN. First, our population was generally older than the older adults in these papers, to the extent that our younger patients would be considered to belong to their ‘older’ groups. One interpretation of these data could be that the relationship between age and neural population power is non-linear such that adults in late middle age show the highest power. Second, there may be task effects, since our data were recorded with the patient not performing a cognitive task, while previous reports have all been in the context of active cognitive tasks. Third, our data may show a genuine difference in the effect of aging in the STN compared to the cortex. Finally, the possibility exists that the change in power we observed reflects age-related changes in shape and location of the STN [44]. According to this hypothesis, recording sites at the same stereotaxic coordinates may represent different functional domains, with different neuronal properties, within the STN in different age groups. Future work may elucidate these possibilities.

#### Multi-Unit Activity

MUA is an important measure to identify the entrance of the STN. Both through visual and auditory inspection and by automatic identification algorithms, an increase in neuronal activity, compared to the overlying white matter zone, is one of the most widely used signs of the dorsal border of the STN [45]. Our results show that sedatives, either continuous or discontinued, as well as a number of patient specific characteristics, influence MUA. To our knowledge, no previous studies have investigated the effect of PSA agents on MUA, despite its usefulness as a clinical marker. More research is necessary to define whether these effects are large enough to be clinically relevant.

### 4.3. Limitations

Our study had some limitations. First, it was a retrospective study, involving patients for whom there was no standardized PSA protocol and uncontrolled variability in PSA drug choices and doses. We found that demographic and clinical characteristics did not differ systematically between the groups, therefore these factors could be accounted for with a regression analysis thanks to our large overall sample size. Since no standardized PSA protocol was used we were limited when testing for dose-dependent effects. Second, while we compared our findings to previous reports, the available literature is sparse and characterized by heterogeneity in PSA protocols, patient characteristics and MER data analysis. Our findings suggest that some differences among previous reports may be accounted for by dose-dependent effects (e.g. on firing rates). Prospective studies with standardized PSA protocols are required to confirm these findings. Also, we did not assess spectral changes in the local field potential. Investigating the effect of PSA on the presence of pathologic oscillations remains unanswered. Another limitation is that this study did not assess the effects of the various agents on intraoperative clinical measures such as tremor. A follow-up study is needed to assess the effects of the various PSA protocols on STN depth and size. Moreover, it is important for clinical practice to address in future studies whether the use of PSA influences clinical outcome of STN DBS.

## 5. Conclusions

When administering sedation during DBS electrode implantation, the aim is always to achieve an optimal balance between patient comfort and good quality MER, to allow optimal placement of the probe. Our results showed that PSA influenced neuronal properties in the STN and that the DEX effect was dose-dependent. Moreover, patient dependent characteristics influenced MER. Whether these effects are large enough to be clinically relevant was not addressed in this study.

## Figures and Tables

**Figure 1 jcm-09-01229-f001:**
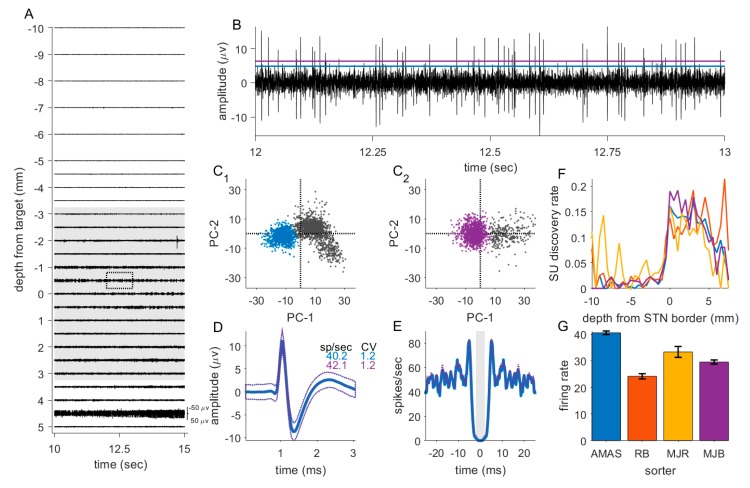
Overview of the sorting process. (**A**) Example recording trajectory of one electrode with 5 s of data shown at each site. Grey shading indicates sites identified as within the subthalamic nucleus (STN). (**B**) Expanded view of data from one recording site (region highlighted by a dashed box in A) with the spike detection threshold set by A.M.A.S. (blue) and M.J.B. (purple); (**C**) clusters of the spike waveforms identified from the example recording by A.M.A.S. and M.J.B.; (**D**) spike waveform (mean: thick lines, thin lines: standard deviation) of the spike sorted from the example recording by A.M.A.S. and M.J.B.; (**E**) Autocorrelation of the spike times from the example recording by A.M.A.S. and M.J.B.; (**F**) The proportion of sites with identified single units (SU)s (discovery rate) was quite similar, indicating that all sorters applied comparable criteria. (**G**) Nevertheless, the mean firing rate of STN neurons differed somewhat between sorters.

**Figure 2 jcm-09-01229-f002:**
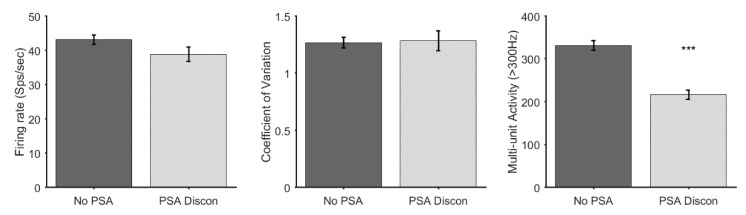
Bar plot and t-test for within awake group analysis for firing rate, defined in spikes per seconds, coefficient of variation and multi-unit activity, defined as activity above 300HZ. Mean and standard error are shown in each bar. *** *p* < 0.001. PSA: procedural sedation and analgesia.

**Figure 3 jcm-09-01229-f003:**
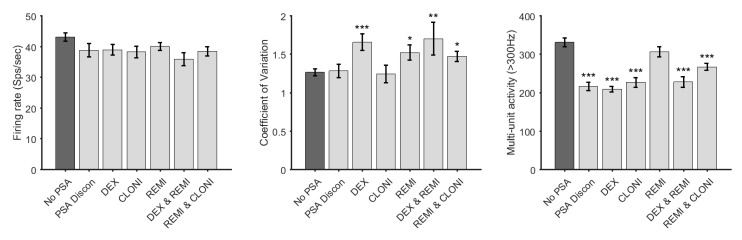
Bar plot (error bars show standard error) and t-test for the No PSA group against each PSA group for firing rate, coefficient of variation and multi-unit activity. * *p* < 0.05, ** *p* < 0.01, *** *p* < 0.001. CLONI: clonidine; DEX: dexmedetomidine; REMI: remifentanil; PSA: procedural sedation and analgesia.

**Figure 4 jcm-09-01229-f004:**
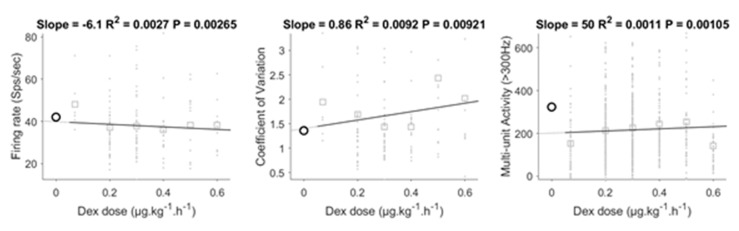
Correlation analysis on the dose of DEX for the electrophysiological measures. Correlations were significant. Data of the no PSA drug group is shown in black for comparison but was excluded from the regression analysis. DEX: dexmedetomidine.

**Table 1 jcm-09-01229-t001:** Demographic and clinical data of all patients.

Group		Patients M/F	Hemispheres (*n*)	Electrodes (*n*)	SU/MUA (*n*)	Age (*y*)	UPDRS III	Dose Range	
Awake	No PSA	15/6	42	149	165/1257	59.4 ± 8.3	38.2 ± 16.5		
	PSA discon	4/3	14	42	51/339	59.3 ± 7.6	42.0 ± 8.5		
Sedation	DEX	7/4	22	70	86/565	60.9 ± 5.9	33.4 ± 12.4	0.07–0.6 µg kg^−1^ h^−1^	
	REMI	8/6	28	99	93/636	58.1 ± 8.0	37.9 ± 14.3	0.02–0.25 µg kg^−1^ min^−1^	
	CLONI	6/2	16	56	41/339	64.8 ± 8.3	35.8 ± 11.9	20–50 µg h^−1^ or 30–150 µg IV in bolus	
	DEX + REMI	4/1	10	35	46/267	65.8 ± 6.5	34.3 ± 7.8	DEX 0.3–0.5 µg kg^−1^ h^−1^	REMI 0.02–0.05 µg kg^−1^ min^−1^
	CLONI + REMI	6/6	24	84	106/679	59.5 ± 7.3	36.4 ± 9.1	CLONI 20 µg h^−1^ or 45–150 µg IV in bolus	REMI 0.01–0.09 µg kg^−1^ min^−1^
**Total**		50/28	156	535	588/4082	60.5 ± 7.7	37.0 ± 12.8		

Values are expressed in mean ± SD. UPDRS III scores are preoperative scores in OFF-state. DEX: dexmedetomidine; CLONI: clonidine; REMI: remifentanil; MUA: multi-unit activity; PSA: procedural sedation and analgesia; SU: single unit; UPDRS III: Unified Parkinson Disease Rating Scale part III; n: number; y: year.

**Table 2 jcm-09-01229-t002:** Standard linear regression model for firing rate, CV and MUA.

Variable		Firing Rate		Coefficient of Variation		Multi-Unit Activity	
		Estimate	*p*-Value	Estimate	*p*-Value	Estimate	*p*-Value
**DEX**	**Yes**	−3.121	0.068	0.313	0.001	−88.814	<0.001
REMI	Yes	−2.105	0.141	0.163	0.040	2.607	0.806
CLONI	Yes	−2.156	0.203	−0.004	0.963	−50.923	0.675
PSA (DISCONTINUED)	Yes	−3.836	0.120	−0.011	0.933	−115.240	<0.001
SEX	Male	0.379	0.808	−0.031	0.720	−2.112	0.855
ONSET SIDE	Ipsilateral (*)	−1.840	0.152	0.100	0.161	−16.353	0.090
HEMISPHERE	Right	1.854	0.150	−0.087	0.222	24.785	0.010
AGE	Years	−0.091	0.314	0.001	0.788	−43.996	<0.001
WEIGHT	Kg	−0.004	0.931	−0.003	0.337	0.452	0.181
UPDRS III		0.068	0.196	0.002	0.428	−0.026	0.946
DISEASE DURATION	Months	−0.002	0.914	−0.002	0.020	−0.018	0.882
STN DEPTH	mm	0.560	0.130	−0.008	0.681	25.024	<0.001
Model data		R-squared: 0.035 Adjusted R-Squared 0.0149 *p*-value = 0.0553		R-squared: 0.0511 Adjusted R-Squared 0.0313 *p*-value = 0.00241		R-squared: 0.0586 Adjusted R-Squared 0.0558 *p*-value = 9.1e^−46^	

In the analysis, we included demographic and clinical variables as sex, onset side (recording site ipsi- or contralateral to the onset side of the disease), hemisphere (right or left), age, weight, UPDRS III pre-operative off medication and disease duration. Estimate indicate the slope of the line, when negative indicates decrease and positive indicates increase. * ipsilateral to body side with onset of disease. CLONI: clonidine; DEX: dexmedetomidine; REMI: remifentanil; PSA: procedural sedation and analgesia; UPDRS III: Unified Parkinson’s Disease Rating Scale part III.

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
