# Peer review of "Influence of Anesthesia and Clinical Variables on the Firing Rate, Coefficient of Variation and Multi-Unit Activity of the Subthalamic Nucleus in Patients with Parkinson’s Disease"

_jcm, 2020, doi:10.3390/jcm9041229_

Round 1

Reviewer 1 Report

The authors performed a retrospective analysis to determine the effects of commonly used PSA agents on MER during DBS surgery. The authors observed a significant reduction in MUA, an increase of the CV and a trend for reduced firing rate by dexmedetomidine. The effect of dexmedetomidine was dose-dependent for all measures. MUA and CV were also influenced by patient-dependent variables. This is a relevant topic in the field of DBS.

The study would benefit from describing the impact o the various agents on intraoperatory clinical measures such as tremor. Although there was no comparison with the most widely used GABAergic agent (Propofol), as a retrospective study, it was well designed and post hoc analyses were well described. 

Author Response

Response to reviewers’ comments

Reviewer 1

The study would benefit from describing the impact of the various agents on intraoperatory clinical measures such as tremor. Although there was no comparison with the most widely used GABAergic agent (Propofol), as a retrospective study, it was well designed and post hoc analyses were well described.

We would like to thank the reviewer for his positive feedback and suggestions. We agree that the study would be even stronger if the clinical effects of the various agents was also provided. However, these were not systematically reported by the neurologists which is an important limitation of this study. We addressed this in the limitations section.

In our complete sample of 80 patients, only two were treated with propofol which, we feel, is too low a sample size to make strong claims. These patients are therefore not included in our analysis.

Reviewer 2 Report

This study seeks to understand the potential impact of sedative / analgesic medications on the neurophysiology of the human subthalamic nucleus (STN) during deep brain stimulation (DBS) implantation surgery. The study is retrospective, but is very well executed within those constraints, and the results are very clearly described and shown.

Some of the notable strengths of this study include the number of patients providing data and the number of recordings, the replication of analyses using single units sorted by different investigators, and the general concordance of results across different statistical approaches.

Overall, there are no major concerns regarding this report. Some considerations are nonetheless provided here:

To the extent that some variables, perhaps age and UPDRS III, for example, might be co-linear, the clarity of the model results might be obscured. The authors could consider demonstrating the non-colinearity of the included variables.

The biggest unanswered question, which the authors readily acknowledge, is the relevance of any observed changes with PSA to clinical practice or to understanding the neurophysiology of the STN. If the former were impacted, one might imagine that this would be evident in the outcomes of patients undergoing DBS implantation with vs. without PSA (e.g., differences in UPDRS on vs. off DBS scores). However, these changes might be sufficiently small that identification of STN is largely not impaired under PSA, but more subtle differences in the information provided by the neurophysiology may be present. If the authors had some metrics for identifying the STN based upon neural activity (which they must at least conceptually, because neural activity is used to determine the STN borders clinically), these might be compared across groups. For example, is the initial depth at which the STN is encountered, or the estimated size of the STN based upon neural activity different depending on PSA? Note that it would be fair for the authors to suggest these questions are best addressed in a subsequent study; these are simply ideas to potentially increase the value or relevance of the current work to clinical practice.

Spectral changes in the local field potential would be interesting to examine across these patient groups, especially with regard to the presence of pathological oscillations (as those are used by some semi-automated systems to detect STN borders). However, as above, this is reasonably deferred to a future report.

(Very) Minor:

Line 157: The threshold should be described as alpha = 0.05, not alpha < 0.05.

Line 179: Data..."were" analyzed.

Author Response

Response to reviewers’ comments

Reviewer 2

#1 To the extent that some variables, perhaps age and UPDRS III, for example, might be co-linear, the clarity of the model results might be obscured. The authors could consider demonstrating the non-colinearity of the included variables.

Variables

Weight

UPDRS III

Disease duration

r

P-value

r

P-value

r

P-value

Age

0.0154

0.8932

-0.0086

0.9403

0.1112

0.3326

Weight

X

0.0387

0.7367

-0.2035

0.0739

UPDRS III

X

X

0.0998

0.3849

Correlation between each of the continuous variables using Pearson correlation is shown in the table above. No significant correlations were found. If requested by the reviewer, we can add this to the supplementary material and methods.

#2 The biggest unanswered question, which the authors readily acknowledge, is the relevance of any observed changes with PSA to clinical practice or to understanding the neurophysiology of the STN. If the former were impacted, one might imagine that this would be evident in the outcomes of patients undergoing DBS implantation with vs. without PSA (e.g., differences in UPDRS on vs. off DBS scores). However, these changes might be sufficiently small that identification of STN is largely not impaired under PSA, but more subtle differences in the information provided by the neurophysiology may be present. If the authors had some metrics for identifying the STN based upon neural activity (which they must at least conceptually, because neural activity is used to determine the STN borders clinically), these might be compared across groups. For example, is the initial depth at which the STN is encountered, or the estimated size of the STN based upon neural activity different depending on PSA? Note that it would be fair for the authors to suggest these questions are best addressed in a subsequent study; these are simply ideas to potentially increase the value or relevance of the current work to clinical practice.

Spectral changes in the local field potential would be interesting to examine across these patient groups, especially with regard to the presence of pathological oscillations (as those are used by some semi-automated systems to detect STN borders). However, as above, this is reasonably deferred to a future report.

We completely agree with the reviewers’ suggestions. We are currently preparing a follow-up paper investigating the questions above in detail where we hope to give answers to these important clinical questions. To emphasize the importance of this, we addressed this in the discussion.

3# (Very) Minor:

Line 157: The threshold should be described as alpha = 0.05, not alpha < 0.05.

Line 179: Data..."were" analyzed.

Adapted accordingly.